# Quantum droplets with particle imbalance in one-dimensional optical lattices

Jofre Vallès-Muns[1,2★], Ivan Morera[2,3],
Grigori E. Astrakharchik[2,4] and Bruno Juliá-Díaz[2,3]

**1** Barcelona Supercomputing Center, Barcelona 08034, Spain
**2** Departament de Física Quàntica i Astrofísica, Facultat de Física,
Universitat de Barcelona, E-08028 Barcelona, Spain
**3** Institut de Ciències del Cosmos, Universitat de Barcelona,
ICCUB, Martí i Franquès 1, E-08028 Barcelona, Spain
**4** Departament de Física, Universitat Politècnica de Catalunya,
Campus Nord B4-B5, E-08034 Barcelona, Spain

★ jofre.valles@bsc.es

## Abstract

We study the formation of particle-imbalanced quantum droplets in a one-dimensional optical lattice containing a binary bosonic mixture at zero temperature. To understand the effects of the imbalance from both the few- and many-body perspectives, we employ density matrix renormalization group (DMRG) simulations and perform the extrapolation to the thermodynamic limit. In contrast to the particle-balanced case, not all bosons are paired, resulting in an interplay between bound states and individual atoms that leads to intriguing phenomena. Quantum droplets manage to sustain a small particle imbalance, resulting in an effective magnetization. However, as the imbalance is further increased, a critical point is eventually crossed, and the droplets start to expel the excess particles while the magnetization in the bulk remains constant. Remarkably, the unpaired particles on top of the quantum droplet effectively form a super Tonks-Girardeau (hard-rod) gas. The expulsion point coincides with the critical density at which the size of the super Tonks-Girardeau gas matches the size of the droplet.

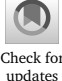

# 1 Introduction

Recently a whole new class of ultra-dilute quantum droplets has been produced in ultracold atomic laboratories with dipolar bosonic atoms [1–3] and bosonic mixtures [4–7]. These quantum droplets originate from a compensation between mean-field and quantum fluctuations [8] and consist of a new type of liquid, which densities can be up to eight orders of magnitude more dilute than liquid helium droplets [9], the only atomic species which naturally remain liquid at zero temperature [10].

Ultracold atomic systems can be subjected to optical lattices created by counter-propagating laser beams [11]. Atoms interact with each other at each site, known as on-site interactions, and can also tunnel through the potential barriers between sites, known as tunneling. Interacting spinless bosons in a high optical lattice are described by the Bose-Hubbard model [12], a model which gained a lot of attention in recent years [13–18]. The use of optical lattices and Feshbach resonances to fine-tune the interactions provides exquisite control over the system and allows the implementation of ideal quantum simulators of Hubbard models, which are ubiquitous in condensed matter.

In particular, ultracold atoms can be trapped to a potential that restricts the movement to only one dimension [19]. One-dimensional geometry allows changing the interaction strength in a much wider range compared to three-dimensional case, due to the suppression of three-body losses [20]. In particular, the coupling constant can be made infinitely repulsive (unitary regime), and a single-component Bose gas acquires a number of fermionic properties [21]. One-dimensional quantum liquids are formed in the regime where the mean-field contribution is on average repulsive [22], differently from the 3D case.

Quantum droplets made of bosonic binary mixtures in a one-dimensional lattice have been studied in the particle-balanced situation [23,24], where the number of atoms of both species is equal. In this work, we consider the fate of quantum droplets in the particle-imbalanced case. First, in Sec. 2 we introduce the model, comment on the used numerical method and briefly review the properties of the balanced case. In Sec. 3, we characterize the effects of particle imbalance in the few-body regime. We find the presence of few-body bound states and characterize them by computing the binding energies and correlation functions. The effects of particle imbalance in the many-body limit are studied in Sec. 4.

## 2 Physical model

We study a binary mixture of bosonic atoms interacting via short-range potential loaded in a deep one-dimensional optical lattice at zero temperature. For a sufficiently high optical lattice, the system properties are well described by the two-component Bose-Hubbard Hamiltonian [13],

$$\hat{H} = -t \sum_i \sum_{\alpha=A,B} \left( \hat{b}_{i,\alpha}^\dagger \hat{b}_{i+1,\alpha} + \text{h.c.} \right) + \frac{U}{2} \sum_i \sum_{\alpha=A,B} \hat{n}_{i,\alpha}(\hat{n}_{i,\alpha}-1) + U_{AB} \sum_i \hat{n}_{i,A} \hat{n}_{i,B} , \qquad (1)$$

where $\hat{b}_{i,\alpha}$ ($\hat{b}_{i,\alpha}^\dagger$) are the annihilation (creation) bosonic operators at site $i = 1,\ldots,L$ for species $\alpha = A,B$; and $\hat{n}_{i,\alpha}$ are their corresponding number operators.

For simplicity, we assume that both species possess the same tunneling strength, $t > 0$, and have equal repulsive intra-species interaction strength, $U > 0$. Throughout the entire work, $t$ is used as the energy scale. We study the case of attractive inter-species interaction, $U_{AB} < 0$, and introduce the dimensionless ratio $r = 1 + U_{AB}/U > 0$.

### 2.1 Numerical method

We use the density matrix renormalization group (DMRG) algorithm to study the ground-state properties numerically. In the DMRG computations used in this work, unless explicitly stated differently, we set a cutoff on the maximum number of bosons of each species per site of $M = 4$ for simulations with sufficiently large interaction strength $U/t$. This gives a local Hilbert space dimension of $d = (M + 1)^2 = 25$. We have explicitly checked that our results are robust with respect to this cutoff, see Appendix A.3 for details on the convergence with the bosonic cutoff $M$. For systems with open boundary conditions, the maximum bond dimension of our DMRG is set to $\chi = 256$ for quantitative results of the density and energy of the system and $\chi = 2048$ when we study correlation functions. For systems with periodic boundary conditions, we use $\chi = 512$ to extract the energy of the system. A study of the convergence of the physical quantities with the bond dimension $\chi$ is presented in Appendix A.2.

### 2.2 Particle-balanced situation

A bosonic binary mixture loaded in a high one-dimensional lattice at zero temperature presents quantum droplets in the particle-balanced situation when the repulsive intra-species interactions are compensated by a comparable attractive inter-species interaction [23, 24]. Here, we provide a brief review of the key aspects of these droplets in the particle-balanced situation, i.e. $N_A = N_B$.

#### 2.2.1 Density profile

Using the DMRG method we are able to obtain the density profile of the ground state which provides important insights into the phase diagram of the system. Specifically, the density profile of a quantum droplet can be well approximated [23] by a symmetrized Fermi function [25],

$$n_{i,\alpha} = \frac{n_M \sinh(R/(2s))}{\cosh(R/(2s)) + \cosh((i - i_M)/s)} , \qquad (2)$$

where $R$ is the size of the droplet, $s$ the typical length scale of the meniscus, $n_M$ is a parameter fixed by the normalization $\sum_i n_{i,\alpha} = N_\alpha$ and $i_M$ is the position of the center of mass,

$$i_M = \frac{\sum_{i=0}^L i \, n_{i,A}}{\sum_{i=0}^L n_{i,A}} . \qquad (3)$$

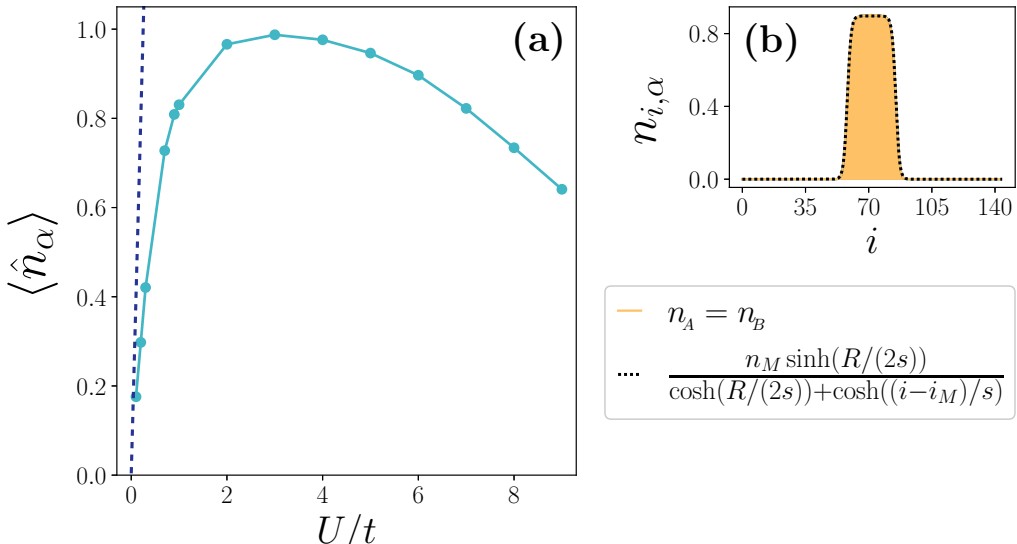

Figure 1: (a) Averaged density of each species in the bulk of the droplet as a function of the interaction strength $U/t$ for $r = 0.15$, the maximum number of bosons per site $M = 6$ and different $L$, ensuring that the droplets fit inside the lattice. (b) Typical density profile of a droplet compared with the corresponding fit using Eq. (2). The droplet in panel (b) is obtained for $U/t = 4$, $L = 144$ and $N_A = N_B = 24$.

In Fig. 1(b), we present a representative density profile for a droplet alongside the fit provided by Eq. (2) for comparison. The value of the central density is an important quantity in the superfluid phase. Its value is determined by calculating the average density within the bulk of the droplet,

$$\langle \hat{n}_\alpha \rangle = \sum_{i=i_M-R/2-4s}^{i_M+R/2+4s} \frac{n_{i,\alpha}}{R+8s}, \tag{4}$$

where $R$ and $s$ are obtained by fitting the densities of the droplet with Eq. (2).

Figure 1(a) shows the evolution of the averaged central density of a droplet as a function of the interaction strength $U/t$ at fixed $r$. For low values of the interaction strength $U/t$ the density tends to the beyond-mean-field (BMF) prediction of the equilibrium density in the continuum, $n_i = 8U/9\pi^2 r^2$ [26]. This is expected since in the limit of low density the distance between the atoms is large compared to the lattice spacing, and discrete description approaches the continuum one. Thus for $U/t \to 0$ we expect to recover BMF results in the continuum. For large $U/t$ the one-dimensional lattice is able to stabilize the density of the droplet and stop the rapidly-growing value of the BMF prediction. This feature shows one of the advantages of the lattice, as lower values in the equilibrium density imply lower three-body losses [27]. It is harder to simulate droplets with low equilibrium density (such as for vanishing or large values of $U/t$) as this requires the use of large lattices in order to obtain converged results. For $U/t = 2/r$ a liquid-gas transition is expected [24] which also corresponds to the threshold for dimer-dimer bound state formation. This transition can be identified as the point where the saturated density of the droplet vanishes at finite $U/t$.

# 3 Few-body systems with particle imbalance

Before studying the effects of particle imbalance in quantum droplets we first address the few-body problem. As in the particle-balanced case [24], the few-body problem offers great insights into the formation of quantum droplets and liquids in the many-body situation. In order to understand the internal structure of the ground state, we perform calculations with varying numbers of particles. This allows us to analyze the different configurations these particles can form, what we refer to as decomposition channels. Each channel represents a potential configuration of interacting subsystems within the overall system, characterized by distinct binding energies and other relevant properties. By examining these channels, we can gain deeper insights into the complex interactions within the system.

## 3.1 Four-particle case

We start by considering a system of four bosons, $N_A + N_B = 4$. In the particle-balanced situation ($N_A = N_B = 2$) the system dimerizes for large values of $U/t$ at fixed and small $r$ [24]. An effective interaction between dimers emerges in this regime. In order to rule out the presence of trimers in the particle-balanced situation we compute their respective binding energies. In Fig. 2(a) we report the energy of two dimers AB and a trimer AAB with a free particle B. The sum of the energy of two dimers is always lower than the sum of the energy of a trimer and the energy of a free particle. Therefore we rule out the formation of trimers in the balanced four-particle case $N_A = N_B = 2$.

## 3.2 Bound states for particle imbalance

After ruling out the presence of trimers in the balanced case we study the formation of bound states with particle imbalance. In Fig. 2(b) we compare the binding energies of the balanced

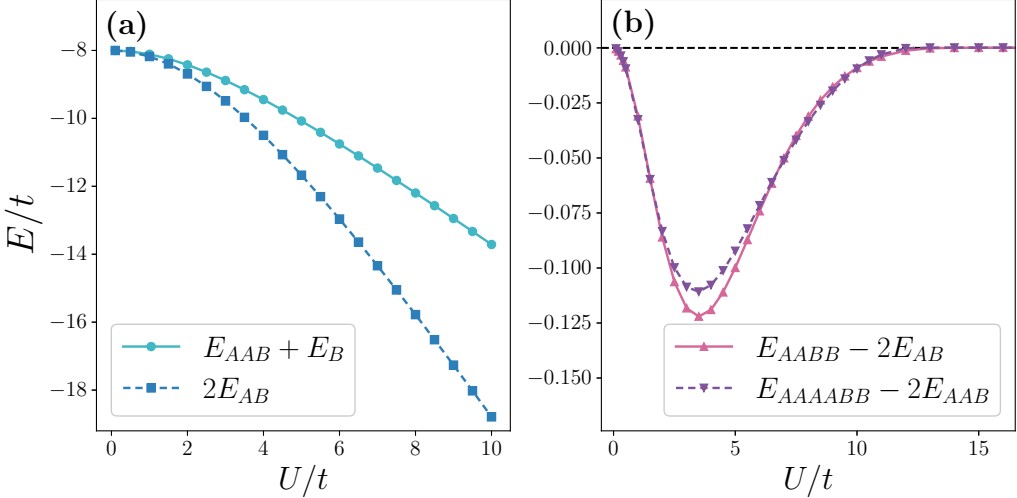

Figure 2: (a) Energy of different particle configurations as a function of the interaction strength $U/t$ for $r = 0.15$ and a lattice of size $L = 200$. Circles show the energy of an AAB trimer and a free B particle while squares represent the energy of two AB dimers. (b) Binding energies as a function of the interaction strength $U/t$ for $r = 0.15$ and $L = 200$. Up (down) triangles show the binding energy of the tetramer (hexamer) obtained by subtracting the energy of two AB dimers (two AAB trimers), respectively.

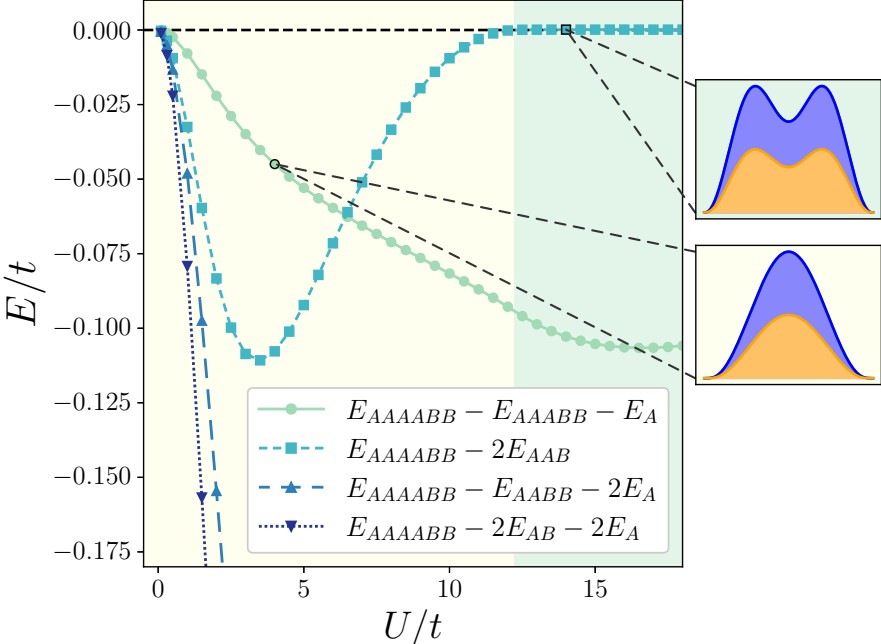

Figure 3: Main plot, the binding energies of the hexamer state ($N_A = 4, N_B = 2$) as a function of the interaction strength $U/t$ for $r = 0.15$ and $L = 200$. The green zone in the background is the region where two trimers are not bound together, this is identified when the binding energy $E_{AAAABB} - 2E_{AAB}$ vanishes. Right panel, characteristic density profiles of the hexamer state calculated for two values of the interaction strengths $U/t$ which are marked with grey dashed lines in the main plot. The blue and orange lines correspond to the density of the A and B species, respectively.

tetramer AABB formed by two dimers and the imbalanced hexamer AAAABB formed by two trimers. We observe that both binding energies exhibit a similar trend for any value of $U/t$ suggesting that quantum droplets may be created in the many-body limit when particle imbalance is present in the system, as we will further explore in Sec. 4. This has to be contrasted with the two-dimensional continuum system where it has been seen that dimers and trimers do not bind at the same coupling strength [28]. Moreover, we have observed that depending on the amount of particle imbalance different large particle composites can be created. We show that the specific structure of bound states depends on the interaction strength $U/t$ and thus, transitions can be observed for a fixed $r$ and a fixed $N_A$ and $N_B$. This can be inferred from Fig. 3, where we study different decomposition channels of the AAAABB hexamer and their respective binding energies. A multiparticle composite is more likely to break when it exhibits a decomposition channel with a small binding energy. In particular, when the binding energy of a decomposition channel is zero the system fully decomposes into the respective particle composites. We find that the hexamer is bound for $U/t \lesssim 12$ at $r = 0.15$ with the most likely decomposition channel being an AAABB pentamer and a single A atom for $U/t \lesssim 6.5$ and two trimers for $U/t \gtrsim 6.5$. The binding energy associated with the decomposition into two trimers becomes zero for hexamer $U/t \gtrsim 12$ indicating that the hexamer fully decomposes into two trimers. The decomposition of a hexamer into two trimers is also reflected in the formation of two bumps in the density profiles denoting the physical separation of the two trimers. With this large variety of bound states in the few-body case, one expects to find intriguing many-body phases associated with the self-organization of these multiparticle composites.

We now study the effect of particle imbalance by changing the number of B particles while keeping fixed the number of A particles. To do so, we start with the balanced configuration

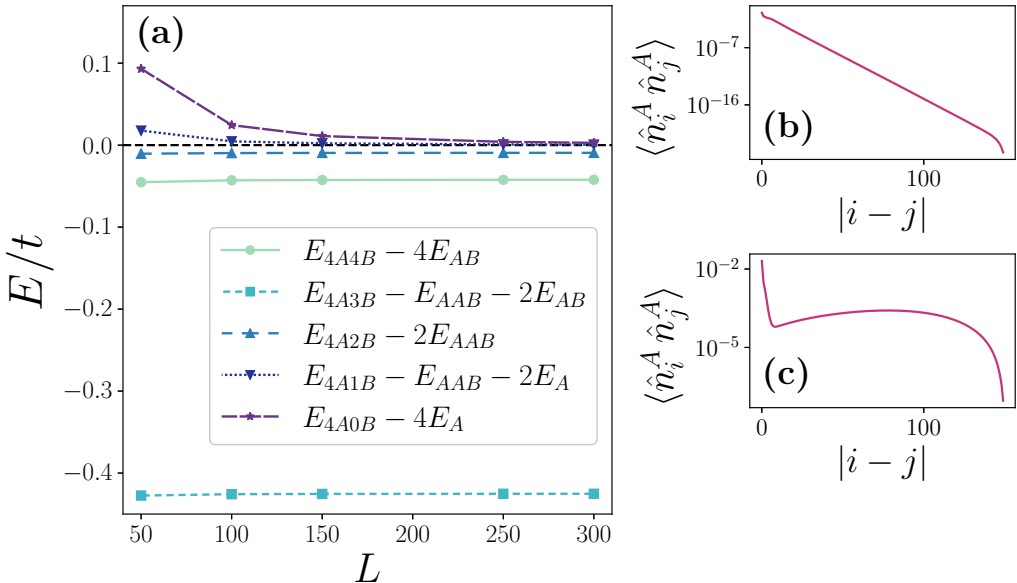

Figure 4: (a) Main decomposition binding energies as a function of the lattice size $L$. Panels (b) and (c), the correlator $\langle \hat{n}_i^A \hat{n}_j^A \rangle$ with $i$ fixed in the middle of the lattice and $j$ scans from $j = i$ to the end of the lattice. In panel (b) the correlator is computed for $N_A = 4, N_B = 2$ and in (c) for $N_A = 4, N_B = 1$. All three panels are obtained for $U/t = 8$ and $r = 0.15$ and both panels (b) and (c) are obtained for $L = 300$.

$N_A = N_B = 4$ and remove B particles. We compute the respective binding energies of all decomposition channels for each imbalance case, from $N_A = N_B = 4$ to $N_A = 4, N_B = 0$. Then, we determine the decomposition channel with the smallest binding energy in each situation. In Fig. 4(a) we plot the binding energy of the most favorable decomposition channel for each imbalance situation as a function of system size. For small imbalances $N_A - N_B \leq 2$ the system is able to bind all particles while for $N_A - N_B > 2$ the system fully decomposes into smaller composites. Thus we conclude that different bound states appear for different imbalances.

Furthermore, to elucidate the internal structure of the system we compute the correlation function $\langle \hat{n}_i^A \hat{n}_j^A \rangle$ shown in Fig. 4(b) and (c). In a bound state, there is an exponential decay of the correlator $\langle \hat{n}_i^A \hat{n}_j^A \rangle \propto \exp(-x/l)$ while the binding energy $E_B \propto -\hbar^2/(ml^2)$ [29], where $m$ is related through the lattice parameters with $m = \hbar^2/(2tl^2)$ [24]. We numerically confirm good agreement with both results. However, when A particles get expelled, the correlation function shows instead a two-regime behavior: it decays exponentially at short distances and then shows saturation at large distances, see Fig. 4(c). Therefore, we find that for $N_B \geq 2$, the B particles bind with the other four A particles forming a large composite object. In contrast, for $N_B = 1$ the B particle alone is not able to bind all the A particles, and instead, with two A particles it creates an AAB trimer while the other two A particles are expelled. Thus, the exponential decay at short distances found in Fig. 4(c) can be identified with the presence of a trimer while the saturated value at large distances is given by the concentration of the expelled particles in a finite-size box and vanishes in the thermodynamic limit, which was numerically confirmed by increasing the lattice size.

# 4 Ground state properties in the particle-imbalanced situation

In the previous section, we have shown that particle imbalance leads to the formation of multiple bound states in the few-body limit. We now show how the presence of these bound states affects the many-body properties of the system. By employing the DMRG method we are able to study systems with fairly large particle numbers and system sizes, sufficiently large for studying the transition from the few-body to the many-body regime.

In the following, we quantify the particle imbalance by means of the dimensionless polarization,

$$z = \frac{N_A - N_B}{N_A + N_B}, \tag{5}$$

where $N_A$ and $N_B$ are the total number of particles for the species A and B, respectively. In a balanced unpolarized system $z = 0$ while in a fully polarized system $|z| = 1$. The imbalance is introduced by removing B particles in the system while fixing the number of A particles.

## 4.1 Particle-imbalanced quantum droplets

Previous beyond-mean-field studies of quantum droplets have shown that pseudo-spin excitations creating particle imbalance are highly energetic and above the particle expulsion threshold, leading to evaporation of the excess of imbalanced particles [8]. In this context, the concept of pseudo-spin can be used to represent the particle number disparity in our spinless bosonic system as an emergent degree of freedom [30,31]. In contrast to the continuum case, here we show that strongly correlated droplets in one-dimensional optical lattices are

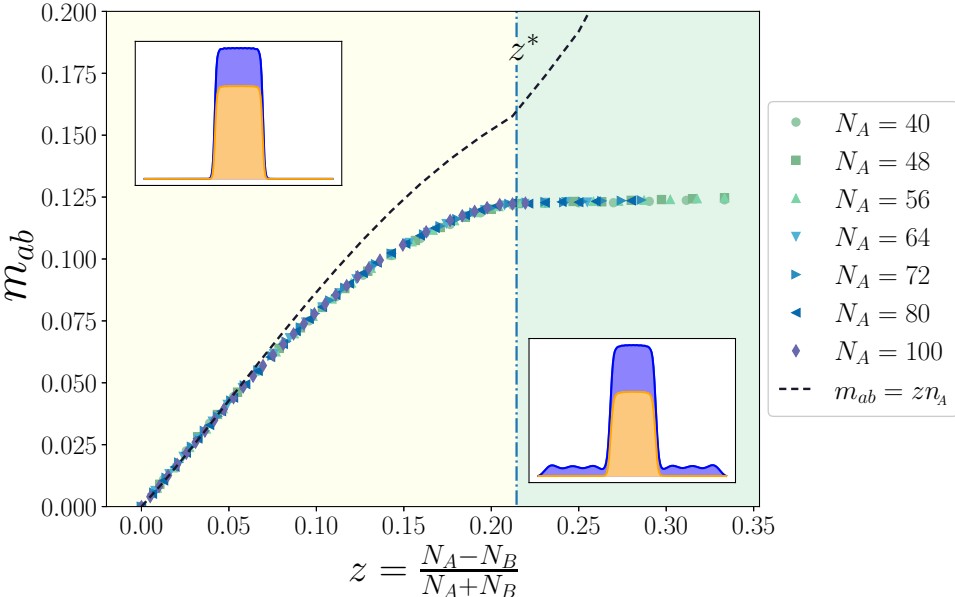

Figure 5: In the main plot, we show the magnetization $m_{ab}$ (defined in Eq. 6) as a function of the imbalance quantity $z$. The dashed line is an analytical approximation for low particle imbalance, and the vertical dashed-dotted line is the value of the critical imbalance $z^*$. In the background, the green region shows where the system expels A particles outside the droplet. In the insets, we show the density profile of components A and B in blue and orange, respectively, in the corresponding $z$ region as a function of the lattice site. This figure was obtained for $U/t = 8$, $r = 0.15$ and $L = 200$. The droplets in the insets are obtained for $N_A = 40$.

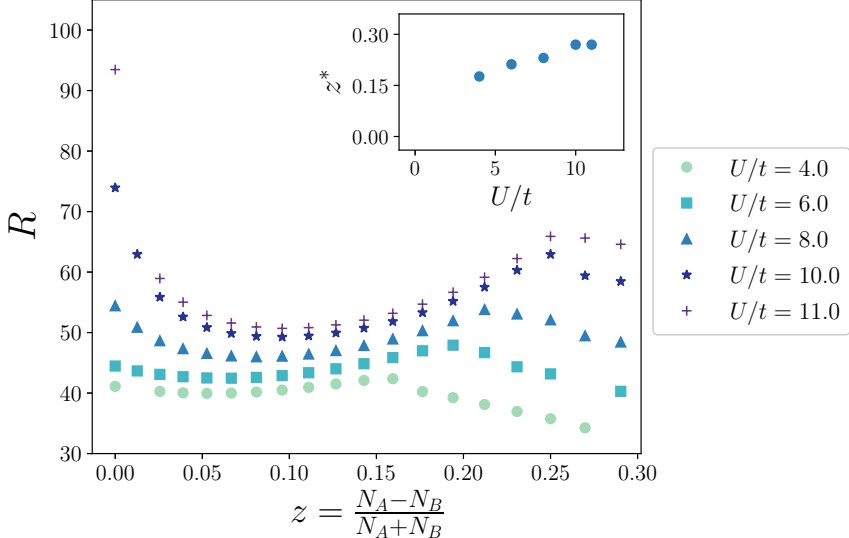

Figure 6: In the main plot, we show the size of the droplets $R$ (obtained from eq. 2) as a function of the imbalance quantity $z$, defined in the text. In the inset, we present the critical imbalance $z^*$ as a function of the interaction strength $U/t$. These results are obtained for $N_A = 40$, $r = 0.15$ and $L = 200$.

robust against a certain amount of imbalance. To characterize the stability we introduce the magnetization quantity,

$$m_{ab} = \frac{\langle \hat{n}_A \rangle - \langle \hat{n}_B \rangle}{2}, \tag{6}$$

where $\langle \hat{n}_A \rangle$ ($\langle \hat{n}_B \rangle$) is the averaged density in the bulk of the droplet for the species A (B), defined in Eq. (4). In Fig. 5 we show the evolution of the magnetization as the particle imbalance $z$ is increased. As the imbalance is augmented, we identify two distinct regimes: the droplet linearly gains magnetization in the bulk for small imbalances while the magnetization in the bulk of the droplet is locked and the excess of particles A are expelled outside the droplet for large imbalances. The expulsion of particles results in a plateau of magnetization as a function of imbalance. The transition between these two regimes occurs at a critical imbalance $z^*$, the specific value of which depends on the interactions in the system. Moreover, these two regimes can be clearly identified by looking at the density profiles, see insets in Fig. 5. Density profiles for $z > z^*$ are characterized by a central droplet with finite magnetization and an outer gas.

In order to take care of possible finite-size corrections we perform simulations for different total number of particles and check if there is a strong dependence on $N$. Instead, we find that the magnetization of the droplets as a function of imbalance $z$ shows a universal behavior for different number of particles, see Fig. 5. Moreover, other physical properties such as the size of the droplet $R$ and its mean density $n$ also exhibit this universal behavior denoting that in our case finite size effects can be safely neglected.

Particle expulsion from the droplet resembles the phenomenon found in the few-body regime discussed in Sec. 3. When particle imbalance is increased the system decomposes into a region of large bound states and a region of non-bound A particles. The critical value of the imbalance can be understood as the point at which the B particles are not able to bind all the other A particles and thus they are expelled.

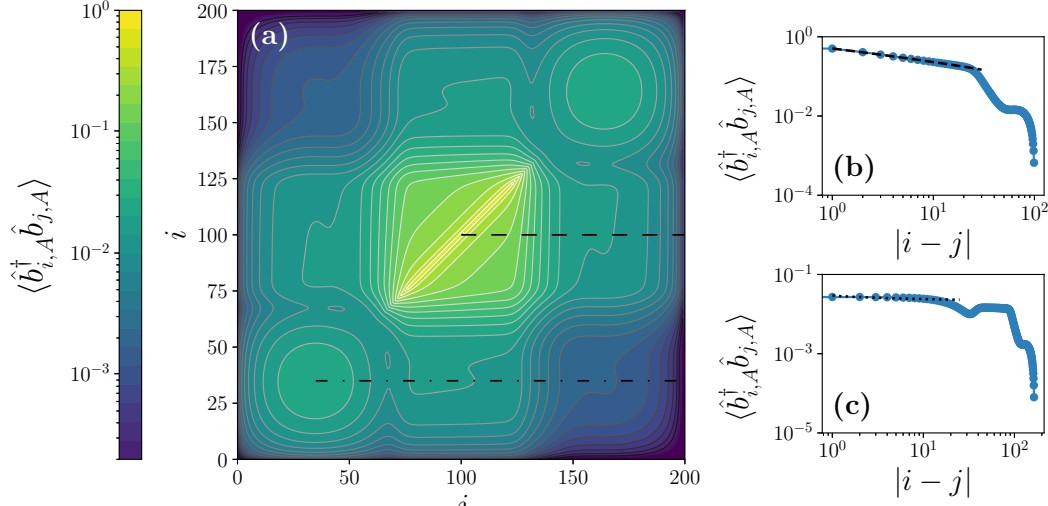

Figure 7: One-body density matrix (OBDM) $\langle \hat{b}^{\dagger}_{i,A} \hat{b}_{j,A} \rangle$ where $i$ and $j$ are lattice sites, applied over a quantum droplet that has two expelled particles outside ($N_A = 40$, $N_B = 24$, $U/t = 8$, $r = 0.15$ and $L = 200$). Both panels (b) and (c) are two cuts in which $i$ is fixed and $j$ goes from $j = i$ to $j = L$. In panel (a) we draw these cuts considered for panels (b) and (c) with a dashed and dashed-dotted line, respectively. In panel (b) the dashed line is a fit inside the droplet region with $\langle \hat{b}^{\dagger}_{i,A} \hat{b}_{j,A} \rangle \propto 1/|i-j|^{\alpha}$, where we extract $\alpha \approx 0.29$. In panel (c) the dotted line is a fit inside the left gas region with $\langle \hat{b}^{\dagger}_{i,A} \hat{b}_{j,A} \rangle \propto 1/\sqrt{|i-j|}$.

## 4.2 Dependence with the interaction strength

In Fig. 6 we represent the size of the droplets $R$ as a function of the particle imbalance quantity $z$. Quantum droplets exhibit a non-monotonous dependence with $z$, and we observe a discontinuity in the derivative of $R$ with respect to $z$, which we identify as the critical value of the imbalance $z^*$ at which expulsion of excess of particles is produced. The size of the droplet decreases with imbalance for $z > z^*$, which we identify as a finite-size effect since the expelled $B$ atoms apply an external pressure to the droplet, thus reducing its size. We expect that the size of the droplet should not depend on the particle imbalance for $z > z^*$ in the thermodynamic limit.

Moreover, we also examine the dependence of the quantum droplets at finite particle imbalance with the interaction strength $U/t$ at a fixed $r$. The size of the droplets increases with the interaction strength $U/t$. Furthermore, the critical value of the imbalance $z^*$ also increases with the on-site interaction $U/t$, see inset in Fig. 6.

## 4.3 Coherence in quantum droplets

We now analyze the coherence properties of an imbalanced quantum droplet that has expelled two particles (one to the left and one to the right) by computing the one-body density matrix (OBDM), see Fig. 7. As we are interested in the coherence between the exterior gas and the droplet, we focus on the OBDM of A species. Remarkably, coherence exists not only inside the droplet but also between the droplet and exterior gas. Figure 7(b) shows the algebraic decay of the OBDM inside the droplet, typical to coherent systems in one dimension. Outside of the droplet, coherence decays even faster. The dashed line in this same panel shows a power-law fit $\rho_{ij} \propto 1/|i-j|^{\alpha}$ with the power exponent equal to $\alpha \approx 0.29$. Figure 7(c) shows the OBDM between the particles in the gas located on the left side and the rest of the system. The

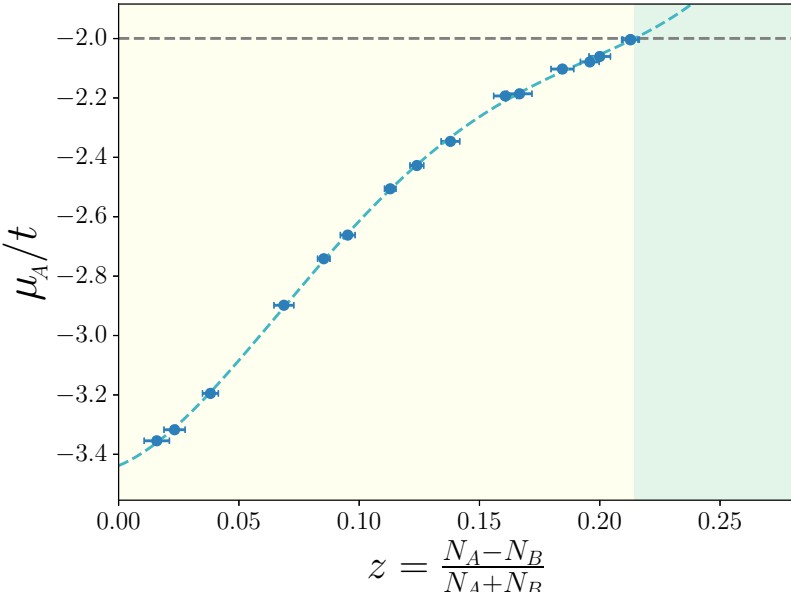

Figure 8: Chemical potential of the A species as a function of the imbalance quantity $z$. The error bars in $z$ come from the discretization of the density obtained from the simulations in open boundary conditions into a finite simulation in periodic boundary conditions. The dashed line is a fit with a quartic function. The green background shows the region where the fit is greater than $-2$. These values are obtained for $U/t = 8$, $r = 0.15$ and $L = 80$.

gas-gas coherence in the same gas region shows a similar decay than the one expected for a Tonks-Girardeau gas $\rho_{ij} \propto 1/\sqrt{|i-j|}$ [32], see dotted line in Fig. 7(c). However, the gas-gas coherence between the left and right gas is significantly suppressed.

## 4.4 Thermodynamic properties

In this subsection we discuss the procedure used for obtaining the thermodynamic properties of droplets as a function of the particle imbalance. Periodic boundary conditions (PBCs) are implemented by adding a long-range coupling between the first and last site of the system. The ground state obtained with PBCs corresponds to a homogeneous solution which for large enough particle number $N = N_A + N_B$ and system size $L$ becomes a good approximation of the thermodynamic limit solution. By exploring the energy of the system as a function of density and imbalance we are able to obtain the full equation of state. Specifically, we fix the total density of the system to match the saturation density obtained in the droplets with open boundary conditions (OBCs) presented in Sec. 4.1. We have explicitly checked that this corresponds to the equilibrium density. Then, we study how the equation of state (EoS) evolves with the imbalance. To extract relevant information of the EoS we compute the chemical potential of the A species,

$$\mu_A = E(N_A, N_B) - E(N_A - 1, N_B), \tag{7}$$

where $E(N_A, N_B)$ is the energy of the homogeneous solution with $N_A$ and $N_B$ particles. The chemical potential $\mu_A$ increases with the particle imbalance $z$, see Fig. 8. At a critical imbalance $z^*$ the chemical potential equals the energy of a single free particle in the Bose-Hubbard model $\mu_A^* = -2t$. At this point, the respective droplet will not be able to sustain an excess of imbalanced particles since their energy becomes lower outside the droplet. Thus, the chemical potential indicates the critical point at which expulsion is expected in finite droplets. The

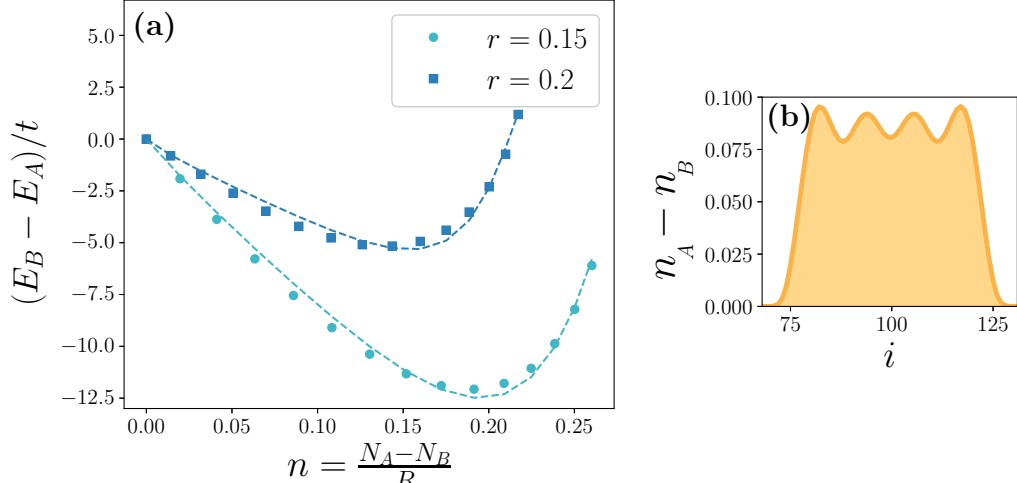

Figure 9: (a) In circles and in squares, $(E_A - E_B)/t$ as a function of $n = (N_A - N_B)/R$ for $r = 0.15$ and $r = 0.2$, respectively. The dashed lines correspond to a fitting with Eq. (9) and considering $J$ and $a$ as free parameters. For $r = 0.15$, $J = 0.888 \pm 0.012$ and $a = 3.596 \pm 0.015$ and for $r = 0.2$, $J = 0.3858 \pm 0.0010$ and $a = 4.72 \pm 0.02$. (b) Density difference in the droplet, $n_A(x) - n_B(x)$, as a function of the lattice site $i$; for a droplet obtained for $N_A = 40$, $N_B = 36$, $U/t = 10$ and $r = 0.15$.

thermodynamic calculation of the critical imbalance $z^*$ agrees very well with the one obtained in finite droplets where expulsion is observed for imbalances $z > z^*$.

### 4.5 Super Tonks-Girardeau gas of the exceeded particles

For small particle imbalance and large interaction strengths $U/t$ we find that the density of exceeded particles, $n_{i,A} - n_{i,B}$ inside the quantum droplet, exhibits $N_A - N_B$ pronounced bumps, see Fig. 9(b). This has to be contrasted with a weakly interacting Bose gas where the density profile is almost flat. Therefore, this indicates that exceeded particles form a highly-correlated state where density-density correlations are enhanced.

To quantify the properties of the highly-correlated gas formed on top of the quantum droplet, we calculate the difference between in energy of the two components $E_A - E_B$, see Fig. 9. We observe that its value vanishes at a critical particle imbalance which resembles the behavior of a lattice Tonks-Girardeau gas but with an effective density,

$$\tilde{n} = \frac{\langle \hat{n}_A \rangle}{1 - \langle \hat{n}_A \rangle (a - 1)}, \tag{8}$$

where $a$ represents the size of the particles measured with respect to the lattice spacing and $\langle \hat{n}_A \rangle$ the mean density of the gas. Given this observation, we compare the energy difference with the energy of a gas of hard rods in a 1D lattice by performing an excluded volume substitution $L \to L(1 - \langle \hat{n}_A \rangle (a - 1))$. This substitution was previously used in the continuum to obtain the energy of the super Tonks-Girardeau (sTG) gas [33]. The same procedure leads to the energy of the sTG in a 1D lattice,

$$E = -2J \frac{\sin(\pi \langle \hat{n}_A \rangle / (1 - \langle \hat{n}_A \rangle (a - 1)))}{\sin(\pi / L(1 - \langle \hat{n}_A \rangle (a - 1)))}. \tag{9}$$

The parameter $J = 1/(2m^*)$ takes into account the effective mass of the exceeded particles propagating on top of the quantum droplet and $a$ gives their effective size. By fitting the free

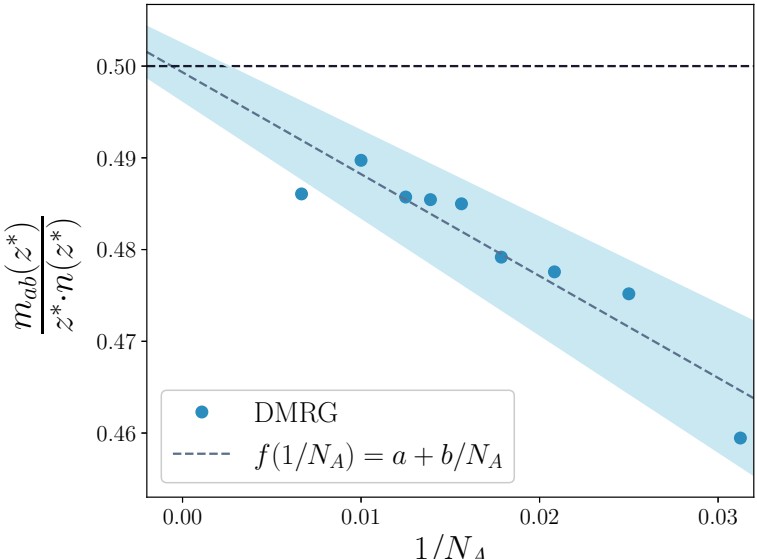

Figure 10: Magnetization normalized by the total density $n = n_A + n_B$ times the imbalance quantity $z$ just before the particle expansion occurs, $z^*$. We fit the obtained values using a function $f(1/N_A) = a + b/N_A$, represented with a grey dashed line in the plot. From this fit we obtain $a = 0.499 \pm 0.03$ and $b = -1.11 \pm 0.17$. The standard deviation of the parameters is displayed with a blue background. These results are obtained for $U/t = 8$, $r = 0.15$ and we choose $L$ ensuring that the droplets fit inside the lattice.

parameters $J$ and $a$ we observe that the energy difference $E_A - E_B$ is well described by the energy of the lattice sTG gas, see Fig. 9. Thus, we conclude that the exceeded particles form an sTG gas on top of the quantum droplet.

Since the exceeded particles form a highly-correlated gas on top of the quantum droplet we can estimate the dependence of the magnetization at small values of particle imbalance. The density of the exceeded gas is given by $\langle \hat{n}_A \rangle = (N_A - N_B)/R$, being $R$ the size of the droplet obtained from Eq. (2). Then, we can write,

$$z = \frac{N_A - N_B}{N_A + N_B} = \frac{R \langle \hat{n}_{\text{diff}} \rangle}{N_A + N_B}, \tag{10}$$

which allows us to express the magnetization as,

$$m_{ab} = \frac{\langle \hat{n}_{\text{diff}} \rangle}{2} = \frac{z(N_A + N_B)}{2R} \simeq z n_A, \tag{11}$$

where in the last step we do a first approximation to the balanced case $(N_A + N_B)/2 \simeq N_A$ and we write $n_A = N_A/R$. Within this approximation, the magnetization is linear in $z$ with a proportionality given by the equilibrium density of the species A. Since the equilibrium density is universal for any number of particles, this approximation of the magnetization is also universal on $z$. In Fig. 5 we show that the magnetization follows this linear dependence for small imbalances $z$.

## 4.6 Bound state insulator

The large bound states observed in the few-body problem suggest that these may have an important role in the many-body scenario. In the low particle imbalance and large interaction

strength regime, we argue that for each B particle removed, a bound state with particle imbalance is formed on top of a dimerized balanced quantum droplet. The large bound states have a size $a$ larger than the lattice spacing. Remarkably, we find that these bound states behave as sTG gas on top of the quantum droplet. By removing more B (increasing imbalance), there is a critical point where the density of bound states becomes commensurate with the droplet size and an insulator is formed $\langle \hat{n}_a \rangle a = 1$. After that, it becomes impossible to fit more bound states inside the droplet for larger imbalances, and thus the excess of A particles is expelled outside the droplet. This provides an estimation of the critical imbalance $z^*$ based on the sTG picture,

$$z^* = \frac{1}{na} \,. \tag{12}$$

The value of the magnetization after expulsion can also be determined,

$$m_{ab}(z^*) = \frac{\langle \hat{n}_A \rangle - \langle \hat{n}_B \rangle}{2} = \frac{1}{2a} \,. \tag{13}$$

With Eq. (13) and (12) we finally obtain,

$$\frac{m_{ab}(z^*)}{z^* \cdot n(z^*)} = \frac{1}{2} \,, \tag{14}$$

which establishes a relation between the value of the magnetization plateau and the critical imbalance at which expulsion starts. This relation is presented in Fig. 10 for a different number of particles. If we extrapolate the values to the thermodynamic limit $N \to \infty$ with a function $f(1/N_A) = c + d/N_A$, where $c$ and $d$ are free parameters, the prediction in Eq. (14) is compatible with numerical results. Furthermore, we also find the size of the bound states obtained from the energy fitting in Eq. (9), the particle imbalance and the magnetization at the critical point $m(z^*)$, given by Eq. (12) and Eq. (13), respectively; have very similar values. This result corroborates our interpretation in terms of sTG gas formed by large bound states.

## 4.7  Magnetic structure within the droplet

We further explore the magnetic correlations within quantum droplets, given that they manifest a finite magnetization in the bulk for imbalances $z < z^*$. For this purpose, we consider two important correlation functions. The pseudo-spin correlator $\langle b_A^\dagger(x) b_B(x) b_A(0) b_B^\dagger(0) \rangle$ measures the extent of magnetic order within the system, indicating how the effective spin states of A and B particles correlate across different locations. Additionally, the pair correlator $\langle b_A^\dagger(x) b_B^\dagger(x) b_A(0) b_B(0) \rangle$ provides us with insights into the tendency of the system to exhibit phase coherence between pairs formed by particles A and B. Figure 11 presents our findings for two homogeneous solutions obtained with PBCs. Figure 11(a) depicts the results in the particle balance, while Fig. 11(b) illustrates the particle-imbalanced case.

In the scenario in which the particle imbalance is present, the pseudo-spin correlator exhibits an exponential decay while the pair correlator decays with a power law. This observation is consistent with the characteristics of the pair superfluid (PSF) phase, which is typified by a gapless region in density and a pseudo-spin sector with a gap. Such behavior, however, changes with the introduction of particle imbalance. In this situation both correlators exhibit an algebraic decay, indicating the closure of the pseudo-spin gap and a transition into the two-superfluid (2SF) phase. This outcome emphasizes the significance of particle imbalance as a crucial variable in the phase space, effectively transitioning the system from the PSF to the 2SF phase. Moreover, we confirm that the imbalance plays a significant role in modulating the magnetic structure of the droplet.

The analytical solution derived in [34], which we use as a benchmark for comparison, is derived within the 2SF region. As such, we can extract the effective Luttinger parameters $K_a$, $K_s$ through a fitting procedure.

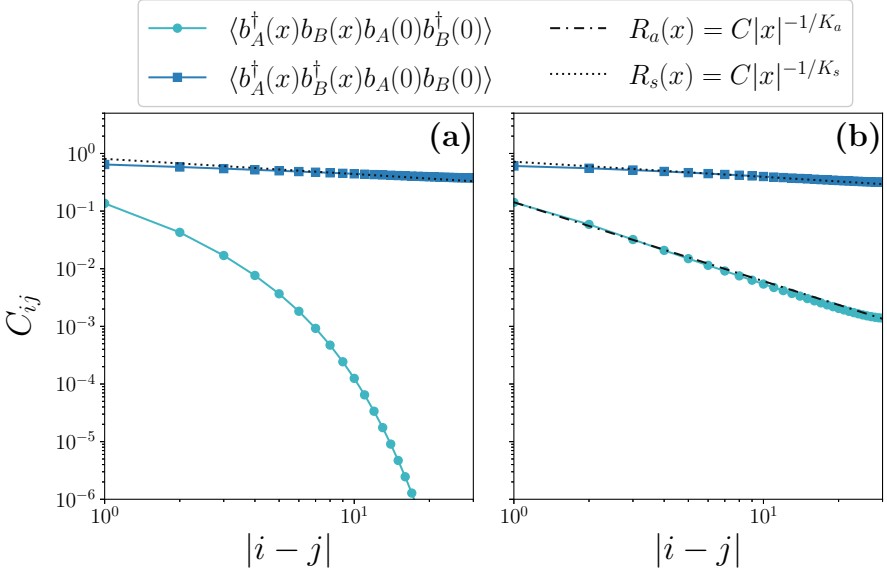

Figure 11: The correlation functions $C_{ij}$ as a function of the separation between sites $|i-j|$, where $i$ and $j$ are lattice sites and $j = 40$ is fixed in the middle of the lattice, for two different homogeneous systems. In panels (a) we display the results in the particle-balanced situation $N_A = N_B = 59$, whereas in (b) we show the results with particle imbalance, $N_A = 65, N_B = 47$. In both situations, these results are obtained for a system with periodic boundary conditions, $L = 80$, $U/t = 8$, $r = 0.15$ and $\chi = 512$. The coefficients obtained from the fit are $K_s = 0.73 \pm 0.01$ and $K_a = 3.9 \pm 0.2$.

## 5  Conclusion

In this work, we have studied the effects of particle imbalance on binary bosonic mixtures at zero temperature in a one-dimensional lattice. We calculated the binding energies in few-body systems of different bound states, which have shown the formation of composite bound states when there is particle imbalance in the system. These composites are not found in the particle-balanced situation. We extracted information about these states by calculating the correlation functions, and we discovered the existence of a critical point at which B particles cannot bind all the other A particles.

In the many-body limit, we studied the magnetization $m_{ab}$. As the imbalance is increased, we identified two distinct regions. In the first one, droplets gain magnetization, resulting in a difference in the density of both species. This produces a system where the densities of both species are proportional to the other. In the second region, the droplet cannot sustain further imbalance. Therefore, it locks the magnetization in the bulk and, for greater particle imbalances, expels particles beyond its boundary. We examined the coherence between the droplet and the exterior gas. The analytical expression of the magnetization presented for low particle imbalance and the relation between the expulsion point and magnetization show very good agreement with numeric results. Using simulations that approximate the thermodynamic limit, we determined the expulsion point by deriving the chemical potential of the majority component in the mixture. We found that the unpaired particles within the droplet effectively form a super Tonks-Girardeau (sTG) gas. Moreover, we discovered that the expulsion point coincides with the critical density at which the size of the sTG gas becomes comparable to the droplet size.

In Appendix A we present the studies of the convergence of the relevant quantities within DMRG simulations. We discuss the importance of these parameters and we explain the scaling of the computational time with these. The notes presented in this Appendix should be interesting for anybody that wants to simulate such systems using DMRG, since often these numeric details are not commonly explained in the literature.

To the best of our knowledge, this is the first time that the effect of particle imbalance has been studied on one-dimensional quantum droplets in an optical lattice. We have shown that droplets are robust against a small particle imbalance and that they are able to gain magnetization. This feature confirms the viability of an experimental implementation, in which more than often a perfectly balanced situation is difficult to achieve. It would be interesting to have a more in-depth study of the correlations between the gas of expelled particles and the entanglement in the system. In addition, it would also be interesting to study the effects of the imbalance in the intra-species interaction strength, $U_{AA} \neq U_{BB}$, since that is often the case in experimental setups.

## Acknowledgments

The authors thank Andrzej Syrwid and Marcin Plodzien for comments concerning the convergence of the method for small values of $r$.

**Funding information** This work has been funded by Grants No. PID2020-114626GB-I00 and PID2020-113565GB-C21 from the MICIN/AEI/10.13039/501100011033 and by the Ministerio de Economia, Industria y Competitividad (MINECO, Spain) under grants No. FIS2017-84114-C2-1-P and No. FIS2017-87534-P. We acknowledge financial support from the Generalitat de Catalunya (Grant 2021 SGR 01411).

## A DMRG convergence

In order to obtain the ground state of the system we employ the Density Matrix Renormalization Group (DMRG) algorithm. This method allows us to obtain the ground state of the system given the number of particles and the system size. At the same time, DMRG sets a number of variational parameters set by the bond dimension $\chi$ and the dimension of the local Hilbert space $d$. To properly obtain reliable physical results we study how the ground state properties depend on the number of these variational parameters. At the same time, our limited classical computational resources force us to reduce the size of the simulations in order to be able to compute them in a feasible time. This balance between these two constraints is what we study in this Appendix.

In each sweep of the DMRG algorithm, we apply an effective Hamiltonian over the Matrix Product State (MPS) updating consecutively one (single-site DMRG) or two sites (two-site DMRG) in order to minimize the energy [35]. Our DMRG computations have been performed using TeNPy [36]. This one uses the two-site DMRG and thus we focus only on this algorithm in the following. The effective Hamiltonian can be written as a matrix of dimensions $\chi_{\max}^2 d^2 \times \chi_{\max}^2 d^2$ [36], where $\chi_{\max}$ is the maximum bond dimension in the two-sites updated and $d$ is the dimension of the Hamiltonian in a single-site. The most computationally expensive part of DMRG is to minimize the energy when the effective Hamiltonian is applied. To do this, we use the Lanczos algorithm [37], which typically converges after a few tensor products that scale $\mathcal{O}\left(\chi_{\max}^3 D d^2 + \chi_{\max}^2 D^2 d^3\right)$, where $D$ is the bond dimension of the Hamiltonian written as a Matrix Product Operator (MPO).

The convergence criteria followed to stop DMRG sweeps is when the relative change in the energy at each tensor update in a sweep is $\Delta E/|E| < -10^{-8}$ and the entropy $\Delta S/S < 10^{-5}$. The computations used in this work have been produced by three different computers. Two of these are desktop computers and the third is a computer cluster. We want to thank Dr. Arnau Rios for allowing us to access this cluster. In Table 1 we detail the main hardware specifications of the three computers.

TeNPy allows parallelizing the code to run on multiple cores. This feature would enable us to take advantage of the vast number of cores that the cluster has. Nevertheless, we have seen that the optimal number of cores in our TeNPy simulations is $2-3$. Since in this work, we focus on the effect of particle imbalance, we need to compute a large number of simulations for a different number of particles between both species. Therefore, we use our computers to simulate a great number of these simulations at once which use $2-3$ cores each.

Another added benefit of working in a CPU cluster is the total amount of RAM size. As an example, a simulation of a droplet with maximum bond dimension $\chi = 4096$ occupies approximately 50 gigabytes of RAM memory. In both desktop computers, the RAM memory is inferior to this number. Thus, we would only be able to compute this simulation using the cluster.

Since we have important but finite computing resources available, a crucial duty is to minimize the size and total time of computations by reducing key parameters in DMRG. This has to be done carefully to obtain meaningful results. In the following subsections, we explain our criteria for choosing two of these parameters: the bond dimension and the maximum number of bosons per site.

## A.1 Initial state configuration

An important practical aspect of the simulations using DMRG is which initial state should be used. Droplets exhibit a non-uniform density profile, since it is flat in the bulk and decays exponentially far away from it. As a result, a simulation initialized with a random state it will generally require a lot of iterations to converge to the ground state. An additional issue that might arise in the ground state simulation of droplets is that they might fragmentate, that is to create more than one droplet in the same lattice. This happens due to the small energy difference between a single droplet and more than one droplet, which is just given by the surface tension energy, and the surface of a droplet in one dimension is reduced to two points.

Therefore, to address these problems and to accelerate the convergence, a good procedure is to start from a state that resembles the ground state. In our situation, we start with a state in which we have two atoms of each species in the center of the lattice, and leave the left and right parts of the lattice empty. As an additional check, we perform simulations for different initial states and check that the energy of a single droplet does not depend on the initial state used.

Table 1: Hardware specifications of the different computers used in this work.

| Computer | CPU model | Number of cores | RAM memory |
|---|---|---|---|
| Desktop computer #1 | Intel® Core™ i5-8400 CPU - 2.80 GHz | 6 | 49.321 GB |
| Desktop computer #2 | Intel® Core™ i5-4430 CPU - 3.00GHz | 4 | 16.456 GB |
| CPU Cluster | Intel® Xeon® Gold 6240R CPU - 2.40 GHz | 96 | 202.35 GB |

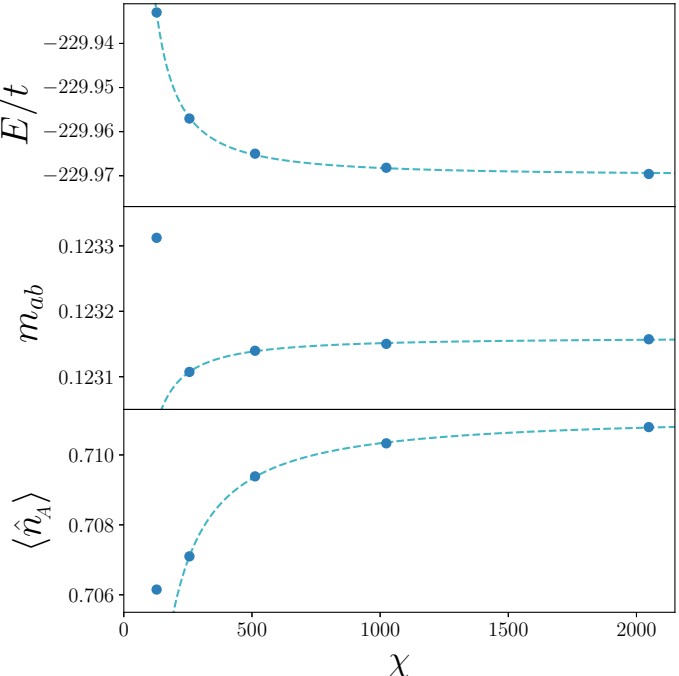

Figure 12: Convergence of different quantities as a function of the maximum bond dimension, $\chi$. We fit each quantity with a function $f(\chi) = a + b/\chi^c$, where $a$, $b$ and $c$ are free parameters. In the second and third panel we exclude the first value to do the fit. Values obtained with a particle-imbalanced droplet for $N_A = 40$, $N_B = 24$, $U/t = 8$, $r = 0.15$, $L = 144$ and $M = 4$.

## A.2 Bond dimension

The bond dimension in an MPS is the dimension of the bond index that connects two following tensors. This quantity can give a measure of the amount of entanglement in the wave function [38]. As we explained, the most computationally expensive part of DMRG scales as $\mathcal{O}\left(\chi_{\max}^3 D d^2 + \chi_{\max}^2 D^2 d^3\right)$. Therefore in DMRG we have to limit the bond dimension up to a predefined value $\chi$ to perform the simulations in a realistic time.

The convergence of the quantities used in this work has to be carefully studied since the value of $\chi$ can have an important role in the results of the simulations. In Fig. 12 we report the convergence of different quantities for a particle-imbalanced droplet with different maximum bond dimension $\chi$. We are able to obtain a prediction to the limit of $\chi \to \infty$ with a fit of the results to a function of $1/\chi$. A crucial quantity studied in this work is the magnetization $m_{ab}$. For $\chi = 256$ the error in the magnetization is on the fourth decimal. We consider that this error is small enough and we choose this value for simulations in which we want to obtain $m_{ab}$.

## A.3 Maximum number of bosons per site

Our system consists of a one-dimensional lattice with $N_A$ and $N_B$ atoms of A and B particles, respectively. Thus, each lattice site can contain from 0 to $N_A + N_B$ atoms. This means that the dimension $d$ of the effective Hamiltonian in one site is $d = (N_A + 1)(N_B + 1)$. Although this would be the exact way to proceed, it is not computationally feasible since the dimension of the local Hamiltonian would be too large. Therefore, we put a cutoff on the maximum number of bosons of each species per site $M$. This sets a maximum dimension of the local Hamiltonian. In this subsection, we study how this value affects the density profiles and the energy of the system.

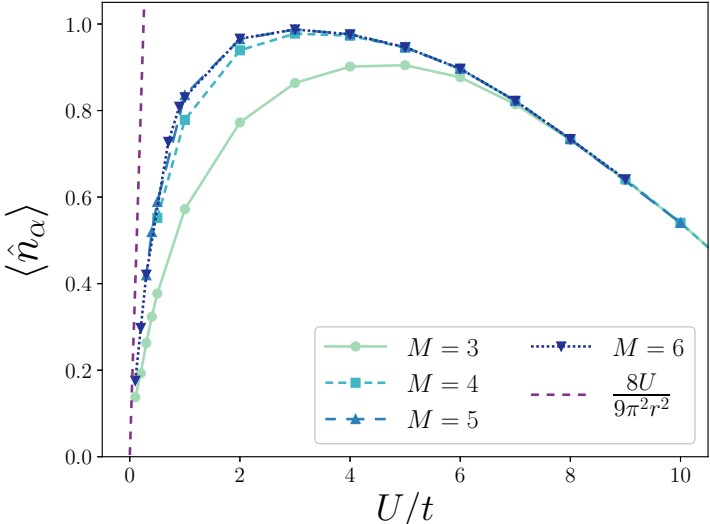

Figure 13: Averaged density of each species in the bulk of the droplet as a function of the interaction strength $U/t$ for $r = 0.15$ and different $L$, ensuring that the droplets fit inside the lattice.

Before introducing particle imbalance, we study the effect of the cutoff $M$ in the balanced situation $N_A = N_B$. In Fig. 13 the mean density of the bulk of a droplet for the balanced situation as a function of the interaction strength $U/t$ is reported. We choose $M = 4$ as the value used for computations in our work since it already shows a good convergence in the

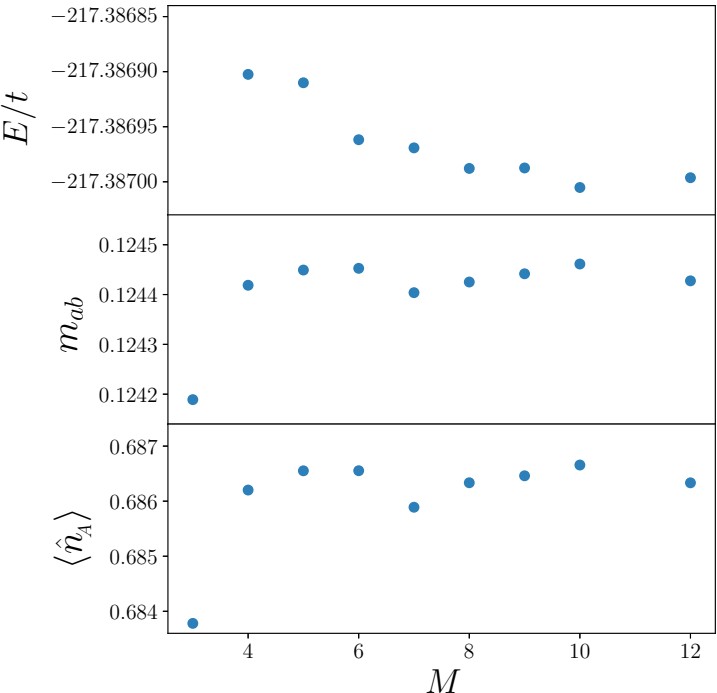

Figure 14: Convergence of different quantities as a function of the cutoff of the maximum number of bosons of each species per site, $M$. Values obtained with a particle-imbalanced droplet for $N_A = 40$, $N_B = 22$, $U/t = 8$, $r = 0.15$, $L = 144$ and $\chi = 256$.

averaged density. Moreover, in our work we focus on the strongly interacting regime, that is the region of sufficiently large $U/t$, which is also the region where the convergence in $M$ is much faster.

Now we consider a system with particle imbalance and we study the effect of the value $M$. In Fig. 14 the convergence of some quantities for different values of $M$ is shown. For $M \leq 4$ the magnetization and density oscillate between the fourth and third decimal, respectively. The energy difference for $M > 4$ is on the fifth decimal, a value equivalent to the convergence criteria of DMRG. Therefore we conclude that for the range of $U/t \in (8, 10)$, $M = 4$ is a sufficient value to obtain valid results.

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
