# Peer review of "Quantum droplets with particle imbalance in one-dimensional optical lattices"

_SciPost Physics, doi:SciPost Phys. 16, 074 (2024)_

## Round 1 · Referee Report · Anonymous (Referee 1) · 2023-8-12

Strengths

Clear and interesting findings.

Well-written manuscript but for some problems mentioned below.

Detailed discussion of the DMRG convergence in the appendix.

Weaknesses

Confusing nomenclature for some concepts, such as magnetization for the density difference between two species of particles without spin.

Apparent violation of the variational principle in some DMRG energy data.

Report

The authors study a generalization of the Bose-Hubbard model for two species of bosons. The ground-state properties are calculated using the density matrix renormalization group (DMRG) method. The numerical results are interpreted on the basis of previous theoretical and numerical works. The present work is motivated by the problem of quantum droplets in optical lattices. The authors focus on the formation and properties of droplets (described by bound states of bosons) when the numbers of bosons of each species are different. They investigate both the limit of few bosons and finite boson densities. The main finding is the observation that droplets are robust against a small particle imbalance. They also claim that unpaired particles within a droplet form a Tonks-Girardeau gas.

Although the manuscript is mostly clear and I believe that the numerical results are probably correct, there are some issues that the authors should correct before I can recommend publication.

1) The authors define a polarization "z" in eq. (5) and a magnetization "m" in eq. (6). They discuss their results in section 4.6 with concepts such as spin correlations, magnetic structure and independent gaps for density and spin excitations. However, the bosonic degrees of freedom do not carry any charge, spin or dipole moment in their model. Although all presented quantities are clearly defined, this nomenclature will be very confusing for most readers. Explain it or change it.

2) The evidence for an effective Tonks-Girardeau gas are rather weak (sections 4.2 and 4.4). Why is this a Tonks-Girardeau gas rather than another type of correlated quantum gas?

3) The accuracy of the DMRG calculations is described in detail in the appendix, which will certainly be helpful for researchers performing similar studies. However, the energy should decreases monotonically with increasing boson cutoff due to the variational principle. The results in the top panel of fig. 14 reveal a convergence problem that must be addressed.

In addition, the manuscript could be improved if the following issues are addressed.

4) Eq. (2) and the formula in fig. 1 correspond to a droplet centered around the origin but the plot in fig. 1 shows a droplet centered around the middle of the lattice.

5) The variable "a" is first introduced as the typical length of the meniscus in eq. (2) then as the particle size in a lattice Tonks-Girardeau gas in eq. (8). Explain the relation between these two lengths or use different symbols if they are unrelated.

6) In the last paragraph of section 3.2, an equation is given for the binding energy in a continuum model with an undefined mass. Explain the relation to the parameters of the lattice model (1).

7) In the text before eq. (8): "... the difference between in energy of the two components ..."

Requested changes

See points 1 to 3 in my reports

  • validity: good
  • significance: good
  • originality: good
  • clarity: good
  • formatting: excellent
  • grammar: good

Author:  Jofre Vallès-Muns  on 2024-01-10  [id 4233]

(in reply to Report 1 on 2023-08-12)

Dear Reviewer,

We are grateful for the detailed evaluations and suggestions, which we believe have significantly contributed to enhancing the quality of our work. In this response, we address the concerns raised by the Report.

1) We thank the reviewers for their insightful comments regarding the terminology used in our manuscript. We acknowledge that the term "spin" might cause confusion, as it traditionally refers to the intrinsic angular momentum of particles, which does not apply to our bosonic spinless system. To improve clarity and precision, we agree to adopt the term "pseudo-spin" throughout our paper to refer to the different components of our bosonic mixture.

2) In Figure 9 in the text (in the new v2 version), we show that the energy difference between both species closely matches the equation of state of a super Tonks-Girardeau (sTG). Also, we show that the difference in the density profile $n_{A}-n_{B}$ shows the same pronounced bumps which are typical of a TG gas. Moreover, we have an analytic interpretation of the magnetization in Eq. (11) that matches our numeric results. We believe that it should be enough to provide evidence of Tonks-Girardeau gas.

3) As nicely stated by Reviewer 2, the variational principle states that the ground-state energy should get lower for a higher boson cut-off, but in our situation we are computing a close upper bound of the ground-state which is not perfectly converged. In the Appendix, we explain that we stop the computation when the relative change in energy in a DMRG sweep is $\Delta E/|E| < -10^{-8}$, in this particular value of the energy, then we have $\Delta E \sim 10^{-6}$, but that does not mean that the energy difference with respect to the ground state is $\sim 10^{-6}$, since this is the energy difference per sweep. Since when we compute the results for different values of bond dimension we start the DMRG with different initial states, then is perfectly possible that in other computations it converges more to the ground state, and some difference in the order of $\sim 10^{-5}$ is very much possible. We thus expect that for more sweeps of the DMRG algorithm, the variational principle would be much more closely followed.

4) Thank you for finding this mistake, you are right that the previous eq. (2) is centered at $i=0$. We have fixed this and updated the text in the new version.

5) You are right that we use two different quantities with the same name. We have updated the text and the typical length of the meniscus is named $s$ instead.

6) Now in subsection 3.2 we explain the relations between the continuum variables and our system.

---

## Round 1 · Referee Report · Anonymous (Referee 2) · 2023-9-11

Strengths

Clear findings, resolves a well-defined problem

Numerics appear to be sound (with one possible exception, see below)

Well-written and quite easy to follow

Weaknesses

I was expecting to see a plot or some discussion of the location of the phase transition as a function of U.

Report

The authors report on 'droplet' formation in the 2-species Bose-Hubbard model with attractive inter-species interaction. The attraction causes the particles to clump together, leading to phase separation with all particles together (in the case of balanced numbers of the two species) with the rest of the lattice being empty. Different to the continuum case, droplets on a lattice support an imbalance in the density of the two species, up to some critical density difference beyond which additional particles 'escape' the droplet. This phenomena was demonstrated in multiple ways, which nicely confirm the picture.

The numerical calculations appear to be solid, for the most part, and I have no doubt that the conclusions are solid.

1 - Figure 5 shows nicely the maximum of the density imbalance ('magnetization') for the fixed value of $U/t = 8$, which appears to be a quite sharp transition, associated with a discontinuity in the second derivative of the magnetization. Are the authors able to state anything interesting about the location of this transition for other values of $U$? Presumably $z^* \rightarrow 0$ for small U, as required for the continuum limit. Are there any analytics for small U, and/or large U?

2- On figure 5, the dashed line represents the quantity $z n_A$, which appears to have a discontinuity in slope at $z^*$, where $n_A$ is stated to be the density of species $A$ in the bulk of the droplet. Why would the slope of $n_A$ be discontinuous at the transition? And why would $z n_A$ increase faster (the curve appears to become slightly quadratic) above the transition? I would expect the opposite - $n_A$ remains approximately constant. Is it a finite-size effect due to the excess $A$ species particles being trapped in the lattice, rather than escaping completely?

3 - On a couple of points raised by the other referee: The nomenclature 'magnetization' for referring to two-species (a 'pseudo-spin' is common in the field, but I agree it is good to introduce this nomenclature better, especially since the paper is otherwise quite clear and accessible to new students.

4- Secondly, on the issue of the energy as a function of the boson cutoff in figure 14, yes in principle the energy is variational and the variational ground-state energy should get lower for a higher boson cutoff, but that doesn't mean that a numerical calculation immediately converges exactly to that variational bound. Indeed, the number of states kept in the calculation, $\chi=256$ is not particularly big, and it is this quantity that sets the scale of the accuracy of the calculation. It is usually not a productive use of CPU time to get a perfectly converged wavefunction for some given bond dimension; if you want a better wavefunction then you are often better off just increasing $\chi$ a bit. For the parameters used in Figure 14, changing $\chi$ by $\pm 1$ changes the energy by $\sim 5\times 10^{-5}$. That is, the range of energy shown on the plot amounts to changing $\chi$ by $\pm 1$. On this scale, the slightly higher energy for $M=12$ versus $M=10$ is a trivial epsilon.

5 - For what its worth, my own calculation for these parameters gives an energy of $E = -217.3873272$ for $M=4$, and $E=E-217.3873397$ for $M=5$. The difference in energy probably comes from 2-site versus 1-site DMRG algorithms. It is known that the 2-site algorithm is not able to produce a perfectly optimal wavefunction for some given bond dimension and I am not at all concerned about this -- for this calculation it is roughly equivalent to increasing $\chi$ by 6 states, and the energy difference from $M=4$ to $M=5$ is consistent with the authors result. However what does worry me a bit is that having determined that $M=4$ is a sufficient boson number cutoff for $\chi=256$, the authors go on to use the same cutoff even when much higher accuracy is required, such as for the $\chi=2048$ calculation in figure 12. My rough estimate suggests that for $\chi=2048$, the cutoff in the boson number becomes significant. As the cutoff in boson number tends to make the state 'more strongly correlated' in some sense, it is not obvious what effect this will have on the long-distance correlations. While I have no doubt that the plateau shown in figure 12 is real, I am not sure whether the dependence on $\chi$ is real or not.

6- Finally, the appendices make a quite good introduction to performing DMRG calculations, and students new to the area could certainly get some benefit from studying this section. But I am wondering if there are some more details that the authors would be willing to share. For these kinds of calculations, where the wavefunction is very non-uniform, getting good convergence of the numerics can be a problem since the energy landscape is so flat. For example, for nearly balanced particle numbers, the droplets form with essentially zero particles elsewhere in the lattice, which makes the calculation quite susceptible to convergence problems, eg forming two separated droplets rather than one droplet. When there is some particle excess, the kinetic energy of the excess particles depends quite sensitively on the size and configuration of the lattice. I expect that the authors went to some lengths to ensure that the droplets in the insets of figure 4 are exactly centred in the lattice, for example. Did the authors do anything special to accelerate the convergence in these cases? A common procedure is to start from a known state that is close to the expected density profile, for example one could start with a weak harmonic trap that localizes the particles in the center of the lattice. If the authors are willing to share these kinds of details, it would also be very helpful for other people attempting similar calculations.

Requested changes

I think the authors need to address point 6 above, and verify that the boson cutoff doesn't affect the long-distance correlations.

The other points would improve the presentation of the paper, but are not essential.

  • validity: high
  • significance: good
  • originality: high
  • clarity: high
  • formatting: excellent
  • grammar: excellent

Author:  Jofre Vallès-Muns  on 2024-01-10  [id 4234]

(in reply to Report 2 on 2023-09-11)

Dear reviewer,

We are grateful for the reviewers' detailed evaluations and insightful suggestions, which we believe have significantly contributed to enhancing the quality of our work. In this response, we will comment the points raised by the report.

1) We have found that this transition also appears for other values of the interactions $U/t$ and $r$. Moreover, for a fixed $r$, the transition is very similar for different values of the interactions $U/t$. We agree that this could be mentioned, and in the new version, we added a new section 4.2 that explains how our results change for different interaction strengths (U/t). We indeed found that $z^{\star}\to 0$ for $U/t \to 0$, but the droplets in $U/t\to 0$ quickly grow in size (see Fig. 1 in the text) and were hard to evaluate without the finite-size effects of the lattice, so we opted not to show the results. For large values of $U/t$ the transition also appears in the droplets, but we also have the problem of too large droplets due to their low equilibrium density (Fig. 1).

2) Remarkably, we found that the slope of both species' equilibrium densities $n_{i}$ definitely changes at the transition $z^{\star}$ (and thus also the total size of the droplet). We agree that this is important and we thus we have added figure 6 in the new version, which shows the size of the droplet ($R$) as a function of the imbalance $z$. The discontinuity in its slope is then related to the discontinuity we find in the slope of $R$. After the transition, we agree that we find a finite-size effect since the expelled B atoms apply external pressure to the droplet, thus reducing its size. We expect that the size of the droplet should not depend on the particle imbalance for $z > z^{∗}$ in the thermodynamic limit.

3) We thank the reviewers for their insightful comments regarding the terminology used in our manuscript. We acknowledge that the term "spin" might cause confusion, as it traditionally refers to the intrinsic angular momentum of particles, which does not apply to our bosonic spinless system. To improve clarity and precision, we agree to adopt the term "pseudo-spin" throughout our paper to refer to the different components of our bosonic mixture.

4) We thank the reviewer for the insightful comments on how our results can still follow the variational principle.

5) We thank the reviewer for this interesting insight. We belive that this point is correct and we do not show this mentioned figure in the new version, since testing again the convergence in $M$ for these large values of $\chi$ will be computationally very expensive. We have updated the text accordingly.

6) We appreciate this comment and we gladly added section A.1 in the Appendix with more information about the initialization of the DMRG algorithm.

---

## Round 2 · Referee Report · Anonymous (Referee 3) · 2024-1-24

Strengths
Clear and interesting findings .
Well-written manuscript.
Detailed discussion of DMRG calculations and their convergence.
Report
The authors have clarified the issues about the presentation and discussion
of their results that were raised in the first round of reports. The manuscript
is well written and organized. The numerical results seem to be correct
and the discussion is now clear. The problem studied by the authors is of
current interest and their work certainly contributes to increasing our
theoretical knowledge of imbalanced bosonic mixtures in optical lattices.
Therefore I recommend publication.

---

## Round 2 · Author Response

List of changes
- As Report 1 suggested, we adopt the term "pseudo-spin" to refer to the degree of freedom in our bosonic mixture.
- As Report 1 mentioned, we fix a typo in Eq. (2).
- As Report 1 suggested, we change the variable name of the typical length scale of the meniscus to "s" instead of "a", since "a" is already used in Eq. (8) for another variable.
- As Report 2 suggested, we remove the previous Fig. 12 since we have not properly checked if the chooses boson cutoff is still valid in the situation of large bond dimension.
- As Report 2 suggested, we added a new subsection (4.2) that studies our results with different interaction strengths (U/t).
- As Report 2 suggested, we added a new section in the appendix (A.1) that explains our procedure to chose a initial state for the DMRG computations.

---

## Round 2 · List of Changes

- As Report 1 suggested, we adopt the term "pseudo-spin" to refer to the degree of freedom in our bosonic mixture.
- As Report 1 mentioned, we fix a typo in Eq. (2).
- As Report 1 suggested, we change the variable name of the typical length scale of the meniscus to "s" instead of "a", since "a" is already used in Eq. (8) for another variable.
- As Report 2 suggested, we remove the previous Fig. 12 since we have not properly checked if the chooses boson cutoff is still valid in the situation of large bond dimension.
- As Report 2 suggested, we added a new subsection (4.2) that studies our results with different interaction strengths (U/t).
- As Report 2 suggested, we added a new section in the appendix (A.1) that explains our procedure to chose a initial state for the DMRG computations.

---

## Editorial Decision

published